# Management of HER2-Positive Breast Cancer for a Young Patient with Visceral Crisis—The Adjuvant Role of Lifestyle Changes

Larisa Maria Badau [1,2], Andrei Dorin Ciocoiu [2], Cristina Marinela Oprean [2,3,4,*], Nusa Alina Segarceanu [2,4], Adelina Gheju [5] and Brigitha Vlaicu [1]

1 Department of Hygiene, "Victor Babeș" University of Medicine and Pharmacy, Bd. Victor Babes No. 16, 300226 Timisoara, Romania; larisa.badau@umft.ro (L.M.B.); vlaicu@umft.ro (B.V.)

2 Department of Oncology—ONCOHELP Hospital Timisoara, Ciprian Porumbescu Street, No. 59, 300239 Timisoara, Romania; andrei.ciocoiu@oncohelp.ro (A.D.C.); nusa.segarceanu@oncohelp.ro (N.A.S.)

3 ANAPATMOL Research Center, "Victor Babeș" University of Medicine and Pharmacy, Eftimie Murgu Sq., No. 2, 300041 Timisoara, Romania

4 Department of Oncology—ONCOMED Outpatient Unit Timisoara, Ciprian Porumbescu Street, No. 59, 300239 Timisoara, Romania

5 Department of Morphopathology, "Victor Babeș" University of Medicine and Pharmacy, Eftimie Murgu Sq., No. 2, 300041 Timisoara, Romania; gheju.adelina@umft.ro

* Correspondence: coprean@yahoo.com; Tel.: +40-745-148-485

**Abstract:** The safety profile and effectiveness of existing anti-HER2-targeted therapies have not been evaluated in patients with breast cancer and visceral crisis. We report the case of a 26-year-old woman who was diagnosed with advanced HER2-positive breast cancer and initially treated with curative intent therapy in a neoadjuvant setting, using Trastuzumab and Pertuzumab in combination with Docetaxel; her cancer recurred two years later, with liver metastases and pulmonary lymphangitic carcinomatosis, causing visceral crisis. Furthermore, the patient's clinical status worsened when she developed respiratory failure, hepatomegaly and a severe hepatocytolysis. Since the patient was free of disease more than six months, we started with Paclitaxel half dose because of the hepatic dysfunction, and we gradually reintroduced Trastuzumab and then Pertuzumab. In the meantime, the patient changed her lifestyle by increasing her consumption of fresh fruits and vegetables and fiber and reducing her intake of processed meat, dairy and sugar. As a result, the patient showed a significant improvement in her respiratory symptoms and liver tests in less than two months. Imaging reevaluation showed partial remission of liver metastases and pulmonary lymphangitic carcinomatosis. She underwent seven months of dual anti-HER2 blockade before relapsing cerebrally. Our results suggest that the sequential combination therapy with Trastuzumab, Pertuzumab and Paclitaxel presented in this study, associated with a healthy lifestyle, may be a good management for recurrent HER2-positive breast cancer with pulmonary visceral crisis and severe liver dysfunction.

**Keywords:** dual anti-HER2 blockade; HER2-positive breast cancer; visceral crisis; healthy lifestyle

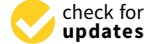



## 1. Introduction

Breast cancer is the most common cancer diagnosed in women, with 2.3 million cases reported in 2020, making it the leading cause of cancer death. In Romania, breast cancer is also a major health problem; approximately 12,000 cases are diagnosed each year, representing 7.2% of the causes of death related to the female population [1,2]. Human epidermal growth factor receptor 2 (HER2) amplification or overexpression is found in approximately 25% of diagnosed breast cancers and has been associated with an increased risk of developing distant metastases in younger women and a decrease in overall and progression-free survival (PFS) [3]. Despite the significant improvement in the survival of HER2-positive

breast cancer patients following the introduction of anti-HER2-targeted therapies, long-term follow-up data indicate that approximately a quarter of patients will relapse [4]. Breast cancer exhibits metastatic propensity to distinct organs, including bones, the lungs and the liver, and up to 50% of breast cancer patients will eventually develop parenchymal brain metastases [5,6]. In the metastatic setting, the combination of Trastuzumab with cytotoxic chemotherapy increased PFS (7.4 months vs. 4.6 months), the objective response rate (50% vs. 32%) and median overall survival (OS) (25 vs. 20 months) [3]. The addition of Pertuzumab to Trastuzumab, along with Docetaxel, has become the standard first-line treatment for HER2-positive metastatic breast cancer (MBC) as a result of PFS and OS improvement, according to the CLEOPATRA study [7]. The triplet also significantly reduced the time to central nervous system relapse by 3.1 months compared to Docetaxel, Trastuzumab and placebo (hazard ratio 0.58, $p = 0.0049$) [7].

About 10–15% of advanced breast cancer will develop visceral crisis, which requires the use of the most rapidly effective therapy, which is not necessarily chemotherapy in all situations [8]. Pulmonary lymphangitic carcinomatosis is a type of visceral crisis frequently caused by breast cancer (33%) [9]. The most common cause of acute liver failure is due to metastases of solid tumors, usually from a breast cancer (30%), which is associated with a poor prognosis, with death occurring within a few days of the clinical presentation [10,11]. Generally, addressing patients with breast cancer and visceral crisis remains a challenge, usually because chemotherapy is often not an option due to liver dysfunction and because the effectiveness of existing anti-HER2-targeted therapies has not been evaluated in randomized trials. In this regard, we report a case of advanced HER2-positive breast cancer whose liver recurrence and pulmonary lymphangitic carcinomatosis caused a life-threatening visceral crisis. In this situation, chemotherapy with Paclitaxel was chosen with the gradual addition of dual anti-HER2 monoclonal blockade with Trastuzumab and Pertuzumab, which led to a significant response by rapidly improving the clinical and biological parameters, as well as the regression of liver metastases and lung lymphangitis. Particularly, correcting the patient's lifestyle behavior during treatment further contributed to this response.

## 2. Case Report

In January 2019, a healthy 26-year-old woman with a normal body weight (Height = 176 cm, Weight = 74 kg, Body Mass Index = 23.89) and who was treated with Levetiracetam for tonic seizures during pregnancy presented to a local clinic for a suspicious palpable mass in her left breast, which had been rapidly growing for the last 2 months. Her medical history was unremarkable: menarche at the age of 12 with regular menses, has used combined oral contraceptive pills for 2 years, after which she had a vaginal birth at the age of 24 and breastfed for 12 months. There was no familial history of cancer.

The physical exam revealed a tumor in the outer quadrants of her left breast measuring 3/2.5 cm which was associated with erythema, skin thickening, purulent mammary secretions and a mobile ipsilateral axillary adenopathy. Magnetic resonance imaging (MRI) revealed a lower outer quadrant tumor with extensions to the upper outer quadrant (Figure 1A,B). The biopsy and the following pathology exam of this ill-defined mass showed an invasive mammary carcinoma of no special type, moderately differentiated (grade 2 of 3) with solid areas of in situ carcinoma of mixed differentiation and extensive comedonecrosis (Figure 2). Further immunohistochemistry showed that the tumor expressed estrogen receptor (ER 70%), progesterone receptor (PR 20%), HER2 overexpression and a Ki-67 index of 40% (luminal subtype, HER2+). An ulterior whole-body computed tomography (CT) highlighted a suspicious 44/40/52 mm tumor in the caudate lobe of the liver and no other metastases. For a better description of the lesion, an MRI was ordered which described it as a focal hyperplasic nodule (HFN). Lab reports and tumor-marker carbohydrate antigen 15-3 were within normal values (CA15-3 = 18.8 U/mL). Correlated with these findings, the tumor was staged as T4bN1M0 stage IIIB according to the AJCC cancer-staging manual (eighth edition).

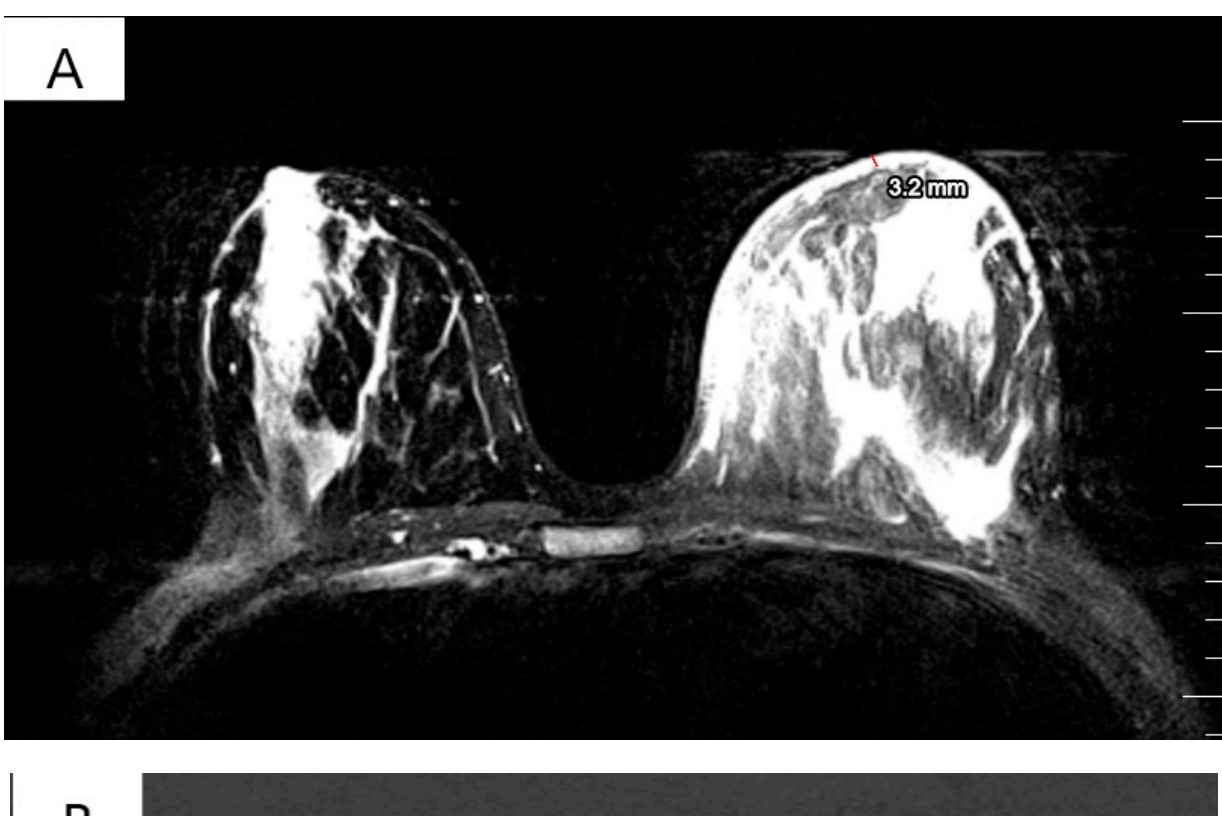

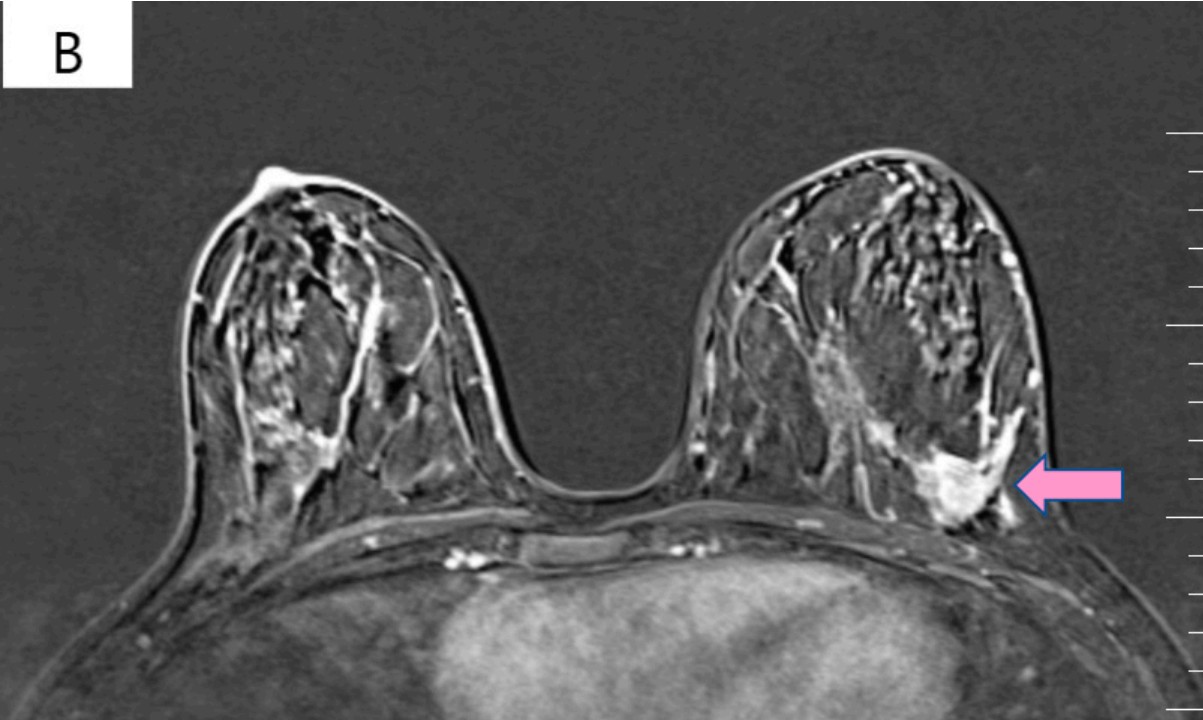

**Figure 1.** (**A**) T2 weighted MRI of left breast demonstrating skin thickening (3.2 mm) associated with hypersignal suggestive of edema. (**B**) T1 weighted MRI showing the regional contrast sample located in the lower outer quadrant extending to the upper outer quadrant (marked by arrows).

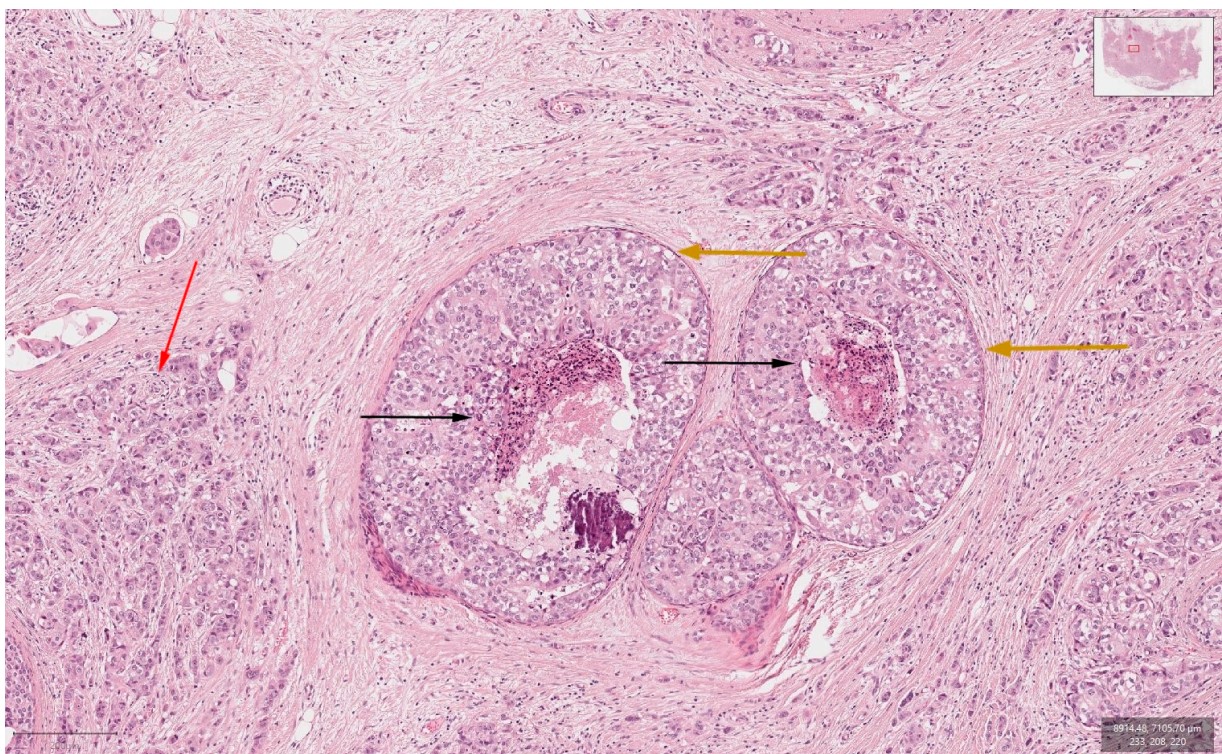

**Figure 2.** Hematoxylin-and-Eosin staining (200× magnification) showing the invasive carcinoma component (marked red arrow), ductal carcinoma in situ component (marked orange arrows) and comedonecrosis (marked black arrows).

In April 2019, the patient began treatment with Docetaxel (75 mg/m$^2$ every 3 weeks), Pertuzumab (840 mg loading dose on cycle 1, followed by 420 mg every 3 weeks), Trastuzumab subcutaneously (600 mg every 3 weeks) and ovarian inhibition with Goserelin 3.6 mg for 28 days. After six cycles of neoadjuvant treatment, her physical examination revealed the remission of the skin thickening and breast tumor, and, respectively, the axillary lymphadenopathy became unpalpable. The treatment was relatively well tolerated, without the occurrence of serious side effects, apart from the expected alopecia. As a result, in August 2019, the patient underwent a left modified radical mastectomy—left breast reconstruction with a tissue expander. The postoperative histopathological examination identified a residual high-grade invasive breast carcinoma (2.3 cm) with intraductal comedonecrosis and cribriform components. Sixteen lymph nodes were examined, and none of them contained metastases (ypT2ypN0cM0). Due to the initial size of the breast tumor and the distance of the tumor from the resection margins (0.4 cm), the patient received adjuvant radiotherapy (two grays daily, 25 fractions) to the left thoracic wall. Afterward, concomitant adjuvant hormone therapy with Anastrozole at 1 mg daily was initiated, and she completed a year of Trastuzumab treatment. Although there was no family aggregation, due to the young age of onset of the disease, we performed BRCA + 16 genes test from the patient's blood, using NGS (next-generation sequencing) and MLPA (Multiplex Ligation-Dependent Probe Amplification) technique (Synlab Diagnosticos Globales, Madrid, Spain) applying Ilumina platform and reagents. Sequencing did not detect any mutations in the BRCA $\frac{1}{2}$ genes. A variant of uncertain significance (class III), NM_000546.5: c.665C > T p. (Pro222Leu), was identified in heterozygous state in the TP 53 gene.

However, 2 years after the initial diagnosis, in February 2021, during the clinical and biological evaluation for an expander replacement, lab reports highlighted abnormal values of the liver enzymes—aspartate aminotransferase (AST) 155 UI/L and alanine aminotransferase (ALT) 114 UI/L. After excluding an infectious or toxic etiology, an abdominal ultrasound revealed the presence of hepatic metastases in the eighth hepatic segment. An

abdominal MRI was immediately requested which showed a liver tumor mass of approximately 9.6/6.6 cm with the aspect of liver metastasis (Figure 3A). Subsequently, chest CT revealed carcinomatous lymphangitis (Figure 4A) and millimeter areas of osteolysis in the T1 and L1 vertebral bodies and the sternal body, without brain metastases on cerebral CT. During workup period, the patient's clinical status worsened by developing symptoms of respiratory failure with dyspnoea at minimal exertion with oxygen saturation (SaO$_2$) of 75% and rapid progressive dry cough. Furthermore, the liver was palpable 6 cm below the costal rim. Hepatic assays were exacerbated by a severe increase in AST 635 UI/L and ALT 309 UI/L (Figure 5), but with normal value of total bilirubin 1.14 mg/dL, alkaline phosphatase 179 U/L and gamma-glutamyl transferase 154 U/L with slightly increased values, albumin 3.01 g/dL, International Normalized Ratio (INR) 1.2. Tumor marker CA 15-3 was also increased (286 U/mL). At that time, the prognosis was extremely reserved.

Due to the patient's clinical deterioration, but also to the severely altered biological values, she was hospitalized to receive emergency treatment to control the symptomatic metastatic disease. Owing to the fact that the patient was free of disease for 10 months, resumption of treatment with Docetaxel and dual anti-HER2 monoclonal blockade with Trastuzumab and Pertuzumab would have been reasonable under adequate liver function. Considering the young age of the patient, we took it into account to reinitiate this treatment, even if are not enough data in the literature about its tolerance and effectiveness in the case of severely compromised liver function. Thus, we chose to start only with Paclitaxel weekly reduced to a half dose (40 mg/m$^2$), due to the lower liver toxicity compared to Docetaxel. On day 8, the transaminases were on a downtrend with the persistence of symptoms but with the appearance of ascites in a moderate amount. However, we increased the dose of Paclitaxel to 60 mg/m$^2$ and added Trastuzumab subcutaneously. The next 2 doses (day 15, 22) of Paclitaxel were given weekly at a dose of 80 mg/m$^2$. After we observed a slight improvement in cough and dyspnoea (SaO$_2$ = 80%) and a decrease in transaminase levels, we switched to a three-weekly protocol of Paclitaxel (175 mg/m$^2$) and introduced Pertuzumab. Starting with cycle two of treatment, the respiratory symptoms subsided, the liver was barely palpable and the remaining ascites was of small quantity.

After six cycles of treatment, an imaging reassessment revealed the partial remission of liver metastases (Figure 3B), HFN with unchanged appearance, and the favorable evolution of carcinomatous lymphangitis lesions (Figure 4B) and bone metastases. In the absence of neurological symptoms, cranial imaging reassessment was not repeated. Laboratory results were within normal limits, and the CA 15-3 decreased to 33.8 U/mL. During this time, the patient changed her lifestyle, she increased her consumption of fresh fruits and vegetables and fiber and reduced her intake of processed meat, dairy and sugar. Moreover, the increase in physical activity led to gradual weight loss, about 9 kg. This time during treatment, she developed an acneiform rash and onychodystrophy, as well as grade I neutropenia and thrombocytopenia. As a result of the good therapeutic response, Paclitaxel was discontinued and hormone therapy with Fulvestrant was added to dual anti-HER2 monoclonal blockade.

Overall, the patient received 7 months of first-line treatment, with a significant clinical benefit. In October 2021, it was discovered that she had multiple brain metastases, following a sudden onset of severe headaches, vomiting and seizures. Laboratory tests remained within normal limits, and the CA 15-3 was 18 U/mL. She received whole-brain radiation therapy (WBRT). She was started on Ado-trastuzumab emtansine (T-DM1) at 3.6 mg/kg intravenously every three weeks. At the time of writing, the patient is on treatment with T-DM1 without toxicity. A new imaging reassessment is scheduled after three cycles of treatment.

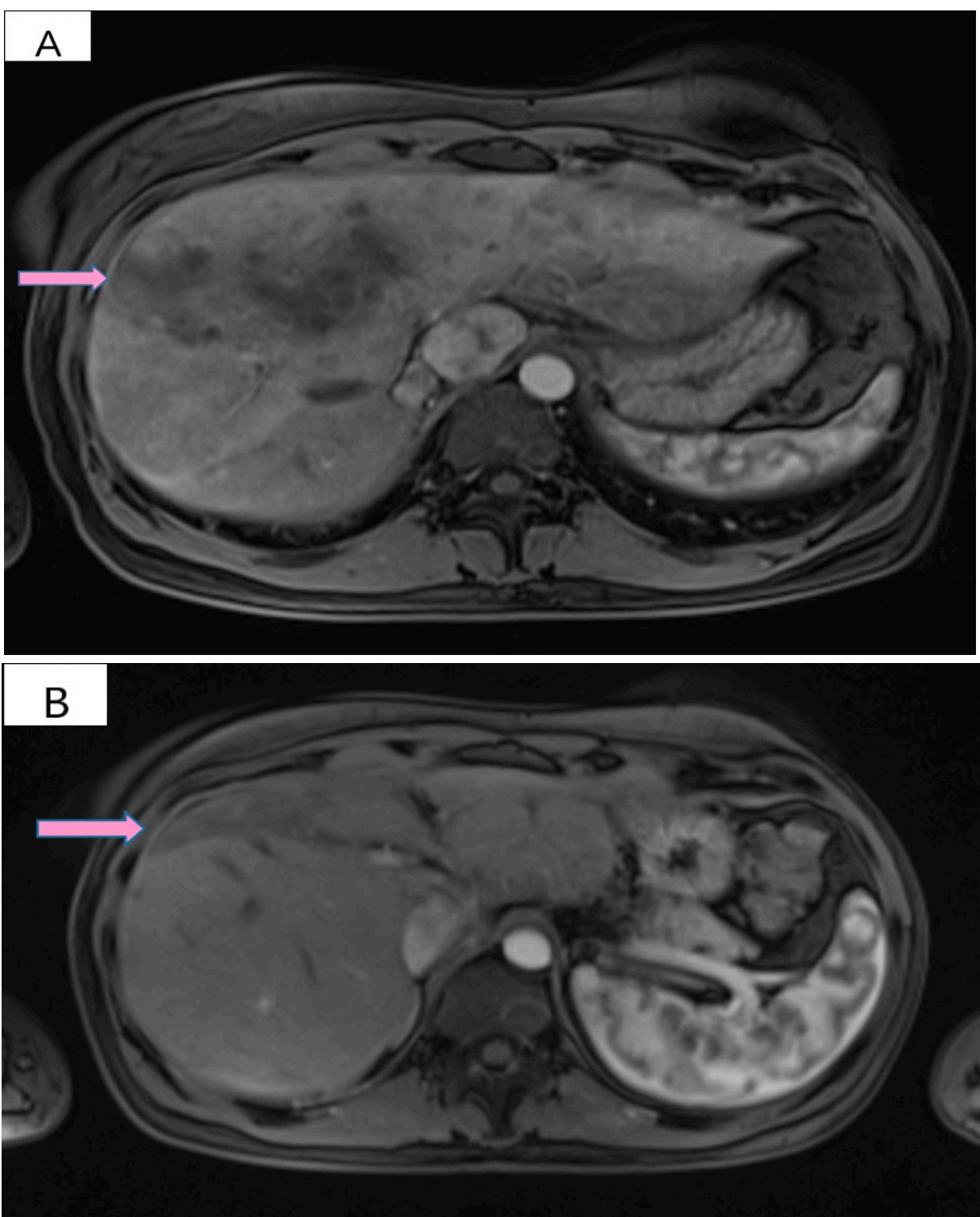

**Figure 3.** (**A**) Pretreatment MRI scan of the abdomen showing a large hypodense mass in the left lobe of the liver—segments VIII and IV (marked by arrows). (**B**) MRI scan showing a dramatic decrease in size of the metastasis (marked by arrows) in the left lobe of the liver after treatment with six cycles of Pertuzumab, Trastuzumab and Paclitaxel.

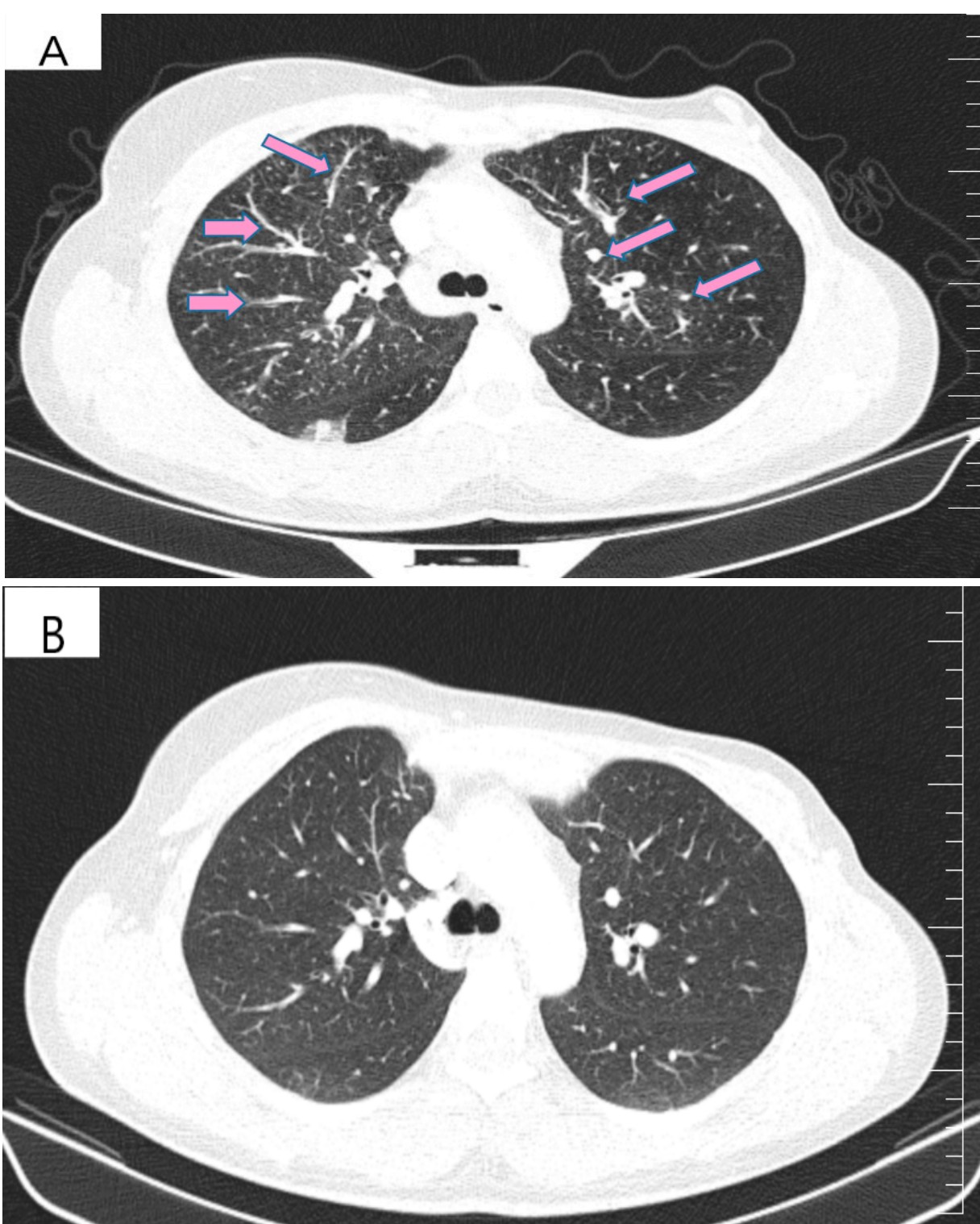

**Figure 4.** (**A**) Pretreatment CT scan of the lungs showing interlobular septal thickening and multiple bilateral peribronchovascular nodular opacities. (**B**) CT scan showing resolution of interlobular septal thickening and remission of most of the pulmonary nodules after treatment with six cycles of Pertuzumab, Trastuzumab and Paclitaxel.

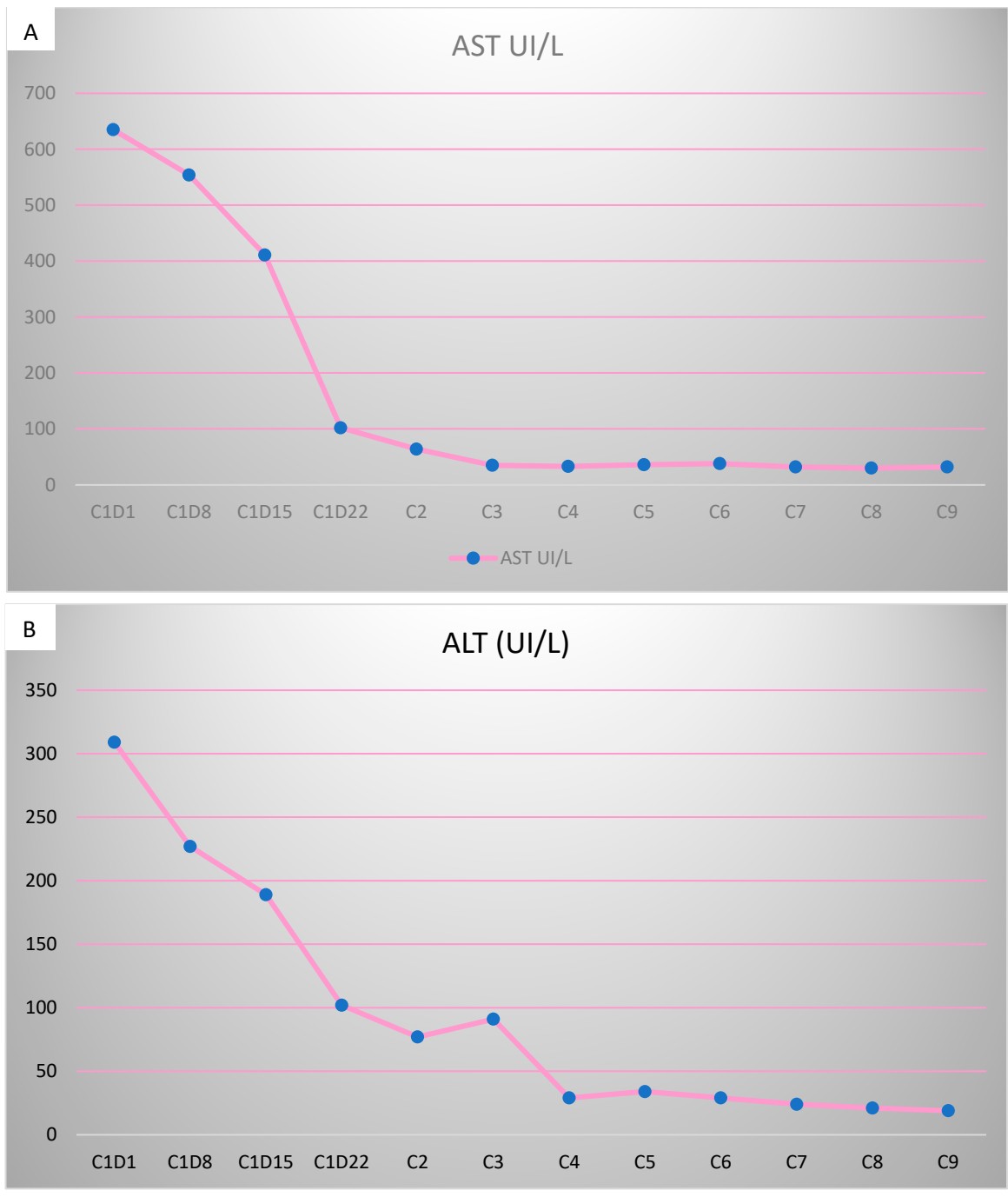

**Figure 5.** Evolution of transaminases during Pertuzumab–Trastuzumab–Paclitaxel therapy. (**A**) Aspartate aminotransferase (AST) values before chemotherapy and after the first administration of Pertuzumab–Trastuzumab–Paclitaxel treatment. (**B**) Alanine aminotransferase (ALT) values during the administration of Pertuzumab–Trastuzumab–Paclitaxel treatment.

## 3. Discussion

Visceral crisis is defined as severe organ dysfunction, as assessed by signs and symptoms, laboratory studies and rapid progression of disease, according to the 5th European School of Oncology (ESO)—European Society for Medical Oncology (ESMO) international consensus guidelines for advanced breast cancer (ABC 5) [8]. Pulmonary lymphangitic carcinomatosis is the malignant infiltration and inflammation of lymphatic vessels and lung interstitium. It prevents proper blood gas exchange and induces resistance to oxygen therapy; therefore, it is considered a visceral crisis. Only about 6–8% of pulmonary metastasis is

caused by lymphangitic carcinomatosis [12]. As with our patient, pulmonary lymphangitic carcinomatosis is usually manifested by dyspnea and unproductive cough. Despite new anticancer therapies developed in recent years, there is still no effective therapeutic strategy to treat pulmonary lymphangitic carcinomatosis. Chemotherapy has been the cornerstone of treatment for patients with carcinomatous lymphangitis; platinum-based protocols have the best response rates [13]. Acute or fulminant hepatic failure caused by solid tumor metastasis is a rare phenomenon, accounting for only 0.44% of all acute hepatic failure cases [14]. Despite the low incidence, when present, it has a mortality rate of up to 90%, even with treatment [14,15]. Given the unfavorable prognosis of the visceral crisis, the main objective is to establish treatment as soon as possible with agents with a tolerable safety profile, thus inducing the best response.

Regarding patients with visceral crisis, the current National Comprehensive Cancer Network (NCCN) treatment guidelines recommend systemic therapy with a single chemotherapeutic agent; combined chemotherapy may be indicated in certain patients with high tumor burden, rapidly progressing disease and visceral crisis [16]. However, choosing a specific chemotherapy or biological agent in the context of organ dysfunction is challenging, especially when there is a continuing decline in performance status. A retrospective study found that the median survival of patients with visceral crisis was 4.7 weeks, and the response to chemotherapy was unsatisfactory [17]. Therefore, the role of chemotherapy in this context is not very clear. HER2 is a transmembrane tyrosine kinase receptor that regulates cellular growth and proliferation in epithelial cells. HER2 amplification is a major driver of tumor cell proliferation and survival in several cancers. HER2-positive breast cancer patients have a poor prognosis and higher recurrence rates, but it is also an opportunity for targeted therapies. Targeting the HER2 pathway provides the most effective therapy, but only in appropriately selected patients [18]. The introduction of anti-HER2 agents has changed the paradigm of HER2-positive breast cancer treatment in the last 2 decades, with monoclonal antibodies, tyrosine kinase inhibitors (TKIs) and antibody-drug conjugates (ADCs) now being the backbone of HER2-positive MBC treatment [19]. The effectiveness of anti-HER2 targeted therapies, such as Trastuzumab and Pertuzumab, has not been evaluated in patients with visceral crisis. Trastuzumab, a recombinant human monoclonal IgG1 antibody, binds to subdomain IV of the HER2 extracellular domain, inhibiting the homodimerization that activates the intracellular signaling pathway and preventing cell proliferation and survival [19]. Pertuzumab is a second-generation anti-HER2 antibody binding to the HER2 extracellular domain II, which prevents dimerization of HER2 with other HER receptors, and together with Trastuzumab, it allows for a more comprehensive signaling blockade [20]. The approval of dual blockade in combination with Docetaxel as first-line therapy is based on the results of the CLEOPATRA trial, due to the median PFS improvement of 18.7 months of the patients in the Pertuzumab group. After approximately 100 months of follow-up, an outstanding improvement of 16.3 months was observed with the addition of Pertuzumab (median OS of 57.1 versus 40.8 months) [7]. The PERUSE study showed the same efficacy when Docetaxel was replaced with Paclitaxel [21].

In breast cancer patients, improved physical activity and diet quality have led to improved OS [22]. Higher well-being is of colossal importance, as it leads to better adherence to oncological treatment and to higher physical activity, which are known to reduce the risk of breast cancer recurrence and mortality [22,23]. In the present case, the potential link between increased consumption of vegetables, fruits and whole grains and decreased consumption of red meat, milk and sweets by the patient may signal an increased ability to tolerate chemotherapy by changing lifestyle behaviors.

Here, we describe our experience with the use of dual anti-HER2 monoclonal blockade with Trastuzumab and Pertuzumab in combination with Paclitaxel in severe hepatic and pulmonary dysfunction secondary to visceral metastasis of a recurrent HER2-positive breast cancer patient. Even though our patient had had this neoadjuvant treatment before, we considered that the recurrence occurred more than 6 months after receiving Trastuzumab, so we were able to reintroduce triple therapy with Trastuzumab, Pertuzumab and a taxane.

However, this was a scenario in which chemotherapy was indicated due to the visceral crisis but with the possibility of worsening hepatocytolysis. Our decision to choose Paclitaxel weekly instead of Docetaxel was influenced by a more appropriate toxicity profile. The major route of excretion of Paclitaxel is hepatic; the pivotal trial recorded grade 1 or 2 liver toxicity in more than 50% of patients receiving Paclitaxel, Trastuzumab and Pertuzumab [21]. As a result, the first dose of Paclitaxel was halved. We witnessed a slight improvement in transaminases that allowed us to increase the dose of Paclitaxel by 25%, and for these reasons, we also added Trastuzumab. Thereafter, Paclitaxel was administered at full dose, with Pertuzumab ulteriorly introduced, with remarkable improvement in transaminases and, in the end, their normalization, as well as improved clinical status. This improvement has allowed us to approach the method of escalating therapy. Urgent initiation of chemotherapy with dual anti-HER2 blockade have been shown to be more effective by rapidly and progressively improving clinical conditions and liver tests.

The literature review highlighted additional case reports in which patients with MBC and severe liver dysfunction were treated by using single-agent chemotherapy with Trastuzumab [24,25]; and only in a single case report was dual anti-HER2 blockade, along with a taxane chemotherapy, used successfully for the same organ dysfunction [26]. This case study used weekly Paclitaxel in combination with Trastuzumab and Pertuzumab in a patient with recurrent breast cancer and acute liver failure, but in contrast to this patient, who had AST 351 UI/L and ALT 144 UI/L, our patient had AST 635 UI/L and ALT 309 UI/L. Macias et al. reported a case of a newly diagnosed patient with metastatic breast cancer and successfully treated liver visceral crisis, whose initial treatment decision was standard therapy with dual anti-HER2 blockade and Docetaxel, but due to the severity of liver dysfunction, platinum-based chemotherapy was chosen [27].

To the best of our knowledge, no other cases of visceral pulmonary crisis successfully treated with dual anti-Her2 blockade along with taxane-based chemotherapy in a first-line setting at the same time with impaired liver function have been reported. In our literature search, we found a case report of a previously Trastuzumab-treated HER2-positive breast cancer relapsed with pulmonary lymphangitis carcinomatosis with benefit to Trastuzumab-emtansine therapy [28].

A proportion of patients with incipient disease will still develop recurrence after neoadjuvant or adjuvant treatment with Trastuzumab, and most metastatic patients will progress during treatment. Following progression during treatment with Trastuzumab (with and/or without Pertuzumab) with a taxane, Trastuzumab emtansine is mainly used based on retrospective exploratory analysis in the EMILIA study [29].

## 4. Conclusions

Even though the effects of therapy in this patient are not fully clarified, our results suggest that combination therapy with Trastuzumab, Pertuzumab and Paclitaxel following the schedule and posology presented in this study may be a good treatment for recurrent HER2-positive breast cancer with visceral crisis and severe liver dysfunction. There are no data on successfully treating hepatic and lung visceral crisis of a breast cancer with dual anti-HER2 monoclonal blockade combined with chemotherapy, and we anticipate that our experience in the future could become a frequent and common practice. Finally, our case shows that the benefits of this sequential therapeutic triad outweigh the risks when it comes to the visceral crisis, and its emergency administration can reverse a potentially fatal situation. It also simply suggests, in addition to conventional medical interventions, the importance of addressing a healthy lifestyle, especially for breast cancer patients suffering from visceral disease. Randomized controlled trials evaluating the impact of diet and physical-activity changes on the treatment outcomes of breast cancer patients are warranted.

**Author Contributions:** L.M.B. participated in the conception, the design of the study, collected the consent, obtained the radiographic and pathologic images and drafted the manuscript. C.M.O. was the attending physician, formatted and submitted the manuscript and reviewed the publication. A.D.C. and N.A.S. collected data and participated to draft the manuscript, and participated in the clinical care and treatment schedule for the patient. A.G. read the pathologic slides, captured the images and helped draft the manuscript. B.V. conceived of the study, participated in its design and coordination and helped to draft the manuscript. All authors have read and agreed to the published version of the manuscript.

**Funding:** This research received no external funding.

**Institutional Review Board Statement:** The study was conducted in accordance with the Declaration of Helsinki and approved by the Ethics Committee of Oncology Center OncoHelp TIMISOARA (approval number 105/20 January 2022).

**Informed Consent Statement:** Written informed consent was obtained from the patient for publication of this case report and any accompanying images.

**Data Availability Statement:** The data presented in this study are available upon request from the corresponding author. The data are not publicly available in order to limit the amount of publicly-available patient personal information, as classified by the European Union General Data Protection Regulation.

**Conflicts of Interest:** The authors declare no conflict of interest.

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
