# Peer review of "Management of HER2-Positive Breast Cancer for a Young Patient with Visceral Crisis—The Adjuvant Role of Lifestyle Changes"

_curroncol, doi:10.3390/curroncol29030154_

Round 1
Reviewer 1 Report
The authors present an interesting case repost and highlight how the combined therapy of Trastuzumab, Pertuzumab and Paclitaxel according to the scheme and posology presented in this study, can be a good treatment for breast cancer.
Minor:
- check the subscript to superscript (line 100, 147, 150, 152)
- convert to italics for Latin terms (line 90)
- standardize versus in vs and in italics (line 51)
- replace the comma with a period (line 77)
Reviewer 2 Report
The manuscript being reviewed is a case study of a young woman with advanced HER2+ breast cancer with recurrence two years later accompanied by visceral crisis. The authors describe how they treated her during the visceral crises, which resolved. This was considered significant because there is not much information on safety and efficacy of HER2+ target therapies during visceral crisis. The patient made significant changes in her diet and exercise. The authors want to conclude that as the result of these changes, she showed significant improvement. While I think there is some level of agreement that such changes in lifestyle may improve a patient’s health, I don’t believe that the science is there to make such a suggestion with this study.
Reviewer 3 Report
In this study, Badau and colleagues described the significance of healthy lifestyle in combination with standard treatments. This case study showed noticeable disease management of HER2-positive breast cancer with pulmonary visceral crisis and severe liver dysfunction when associated with healthy lifestyle. This study is very interesting and suggests useful recommendations for disease management.
The study requires serious proofreading to improve the quality of this manuscript. I highlighted some issues:
Line : 19 , line 24 pacients
Line 18 has (have)
Line 24: Use Since instead of Owing to the fact that
Line 27: Fiber instead of fibre
Line 33: maybe instead of may be
Line 76: remove apparently
Repetition line 80: normal and regular
Line 117: Multiplex-ligation dependent probe amplification not multiple
Line 124: etiology instead of aetiology
Line 127: Liver instead of a liver
Line 133: coastal rim instead of costal rim
minor comments:
Figure 1 not labelled A and B
Legend in Figure 5 is not clear what is the x-axis labelling?
Round 2
Reviewer 2 Report
The authors admit there is not sufficient evidence to support their premise that changing lifestyle behaviors could improve response. However, their way to deal with that is to put an additional sentence in the conclusion. The correction needs to be in the title.
